# Soft Tissue Sarcomas of the Head and Neck Region with Skull Base/Intracranial Invasion: Review of Surgical Outcomes and Multimodal Treatment Strategies: A Retrospective Case Series

Ahmed Habib [1], Idara Edem [1], Diana Bell [2], Shirley Y Su [3], Ehab Y Hanna [3], Michael E. Kupferman [3], Franco DeMonte [1] and Shaan M. Raza [1,*]

1    Department of Neurosurgery, The University of Texas MD Anderson Cancer Center, Houston, TX 77030, USA
2    Department of Pathology, The University of Texas MD Anderson Cancer Center, Houston, TX 77030, USA
3    Department of Head and Neck Surgery, The University of Texas MD Anderson Cancer Center, Houston, TX 77030, USA
*    Correspondence: smraza@mdanderson.edu; Tel.: +713-792-2400; Fax: 713-794-4950

**Abstract:** Soft tissue sarcomas (STS) invading the skull base are rare with little data to guide surgical management. Here we aimed to determine the factors affecting tumor control rates and survival in patients with T4 stage head and neck STS involving the skull base. A retrospective review of STS patients, surgically treated at our institution between 1994 and 2017 was conducted. Variables were collected and assessed against progression-free survival. Tumors were graded using the Fédération Nationale des Centres de Lutte Contre le Cancer (FNCLCC) system. A total of 51 patients (mean age of 35) were included, of whom 17 (33.3%) patients were FNCLCC grade 1, 8 (15. 7%) were FNCLCC grade 2 and 26 (51%) were FNCLCC grade 3. The median PFS was 236.4 months while the 5- and 10-year PFS rates were 44% and 17%, respectively. Recurrence occurred in 17 (33.3%) patients. Local recurrence occurred in 10 (58.8%). Univariate analysis revealed R0 resection had a near-significant impact on tumor control in radiation-naïve patients. Otherwise, prior radiation (HR 6.221, CI 1.236–31.314) and cavernous sinus involvement (HR 14.464, CI 3.326–62.901) were negative predictors of PFS. The most common cause of treatment failure was local recurrence. In T4 stage head and neck STS with skull-base involvement, FNCLCC grade, radiation status, and anatomic spread should be considered in determining the overall treatment strategy.

**Keywords:** skull base; sarcomas; head and neck; multimodal; case series

## 1. Introduction

Sarcomas are broadly categorized into bony and soft tissue origins. STS further encompass a heterogeneous group of tumors of mesenchymal lineage classified based on tissue of origin [1]. Given the numerous tumor subtypes and their relative rarity, they have often been studied collectively under the STS umbrella in the largest clinical studies published to date, irrespective of disease site.

Historically, head and neck STS were staged according to the AJCC Staging System criteria developed for extremity and trunk sites where size > 5 cm was the primary anatomic consideration in local staging and demonstrated to be predictive of outcomes in head and neck sarcomas, specifically [2–4]. Recognizing considerations unique to the head and neck, and the limitations of tumor size in T-staging, a site-specific staging system was adopted by the AJCC in its eighth edition [5]. Notably, invasion into local structures (i.e., skull base) is the primary criteria for T4 staging and several studies have demonstrated this to be significantly predictive of poor long-term outcomes for head and neck STS, regardless of tissue of origin [6,7]. Advanced T-staging is considered a driving factor in guidelines regarding resectability and determining (neo)adjuvant treatment strategies. Beyond TNM staging, due to their histological spectrum, STS are also biologically graded

using the Fédération Nationale des Centres de Lutte Contre le Cancer (FNCLCC) grading system [8] which considers three factors: tumor differentiation, mitotic activity, and extent of necrosis. This grading system aids in classifying a spectrum of tumor subtypes based on biological behavior in a means that correlates with tumor control rates and survival [1]. A combination of the TNM and grading information is then utilized to guide the overall multimodal treatment strategy.

Several retrospective studies to date have demonstrated the role of surgery and the impact of margin status in head and neck STS cohorts spanning different T-stages [2–4]. While there have been additional studies reviewing sarcoma outcomes in the skull base, none to date have focused purely on STS [9,10]. Given the implications of T4 staging and the challenges in their surgical management, along with the biological considerations specific to STS, we analyzed our experience with STS with skull base invasion to better understand the role of surgery in this high-risk subgroup in the context of multimodal treatment strategies.

## 2. Materials and Methods

A retrospective review of the prospectively collected data in the Brain and Spine institutional database was performed. This was done using a protocol not requiring patient consent. The Institutional Review Boards committee in our institution reviewed the study and deemed it in compliance with institutional regulations regarding the study of human subjects. The protocol was institutional review-board approved and met all HIPPA standards. Inclusion criteria for selection were histopathologic confirmation of soft tissue sarcoma, radiographic confirmation of skull base involvement, surgical management between 1994 and 2017, as well as evidence of clinical and radiographic follow-up at our institution. During this time period, 60 surgically treated patients were identified and of these, those with local disease without systemic or nodal metastases were included in this cohort study (51 patients). Tumor- and treatment-related factors were assessed. Anatomic involvement was determined, based on preoperative MR imaging. The extent of resection was based on a comparison of preoperative and postoperative imaging along with the final histopathologic review of soft tissue margins taken during surgery. The extent of resection was classified as follows: R0—gross total resection with negative microscopic margins, R1—gross total resection with positive/unclear microscopic margins, and R2—subtotal resection. All tissue samples were reviewed at the time of surgery by pathologists experienced in the evaluation of soft tissue sarcomas. All tissue samples underwent repeat histologic review for this manuscript in order to confirm the FNCLCC grading. Given the histologic heterogeneity in this disease type, and in line with large published series reviewing STS outcomes at other disease sites, the cohort were categorized according to FNCLCC grade. The primary outcome was progression-free survival. PFS was defined as the time between surgical intervention and radiographic demonstration of either local or distant disease progression.

All variables were assessed with frequency distributions and summary statistics. Correlation between variables was assessed using the paired two-tailed t-test for continuous variables and the chi-squared test for categorical variables. Kaplan-Meier estimates of PFS were performed and the survival curves were compared using the log-rank test. Cox proportional hazards model was used to identify predictors of PFS. A *p*-value < 0.05 was considered significant. All analyses were performed using SPSS (IBM Inc., Armonk, NY, USA, 2017). This case series has been reported in line with the PROCESS Guideline 11.

## 3. Results

### 3.1. Cohort Characteristics

Fifty-one patients were included in this study cohort with a mean age at the time of treatment of 35.4 years. Regarding the status of disease at presentation to our institution, 25.5% were patients who were newly diagnosed, 27.5% had persistent disease after recent treatment and 47.1% had recurrent disease (Table 1). A prior biopsy had been performed

in 47.1% of patients, prior surgical resection had been performed in 62.7%, 33.3% had received prior radiation treatment and 21.6% had received prior chemotherapy. The median KPS at the index surgery was 90. The median overall survival for the whole cohort was 94.53 months and was 170.7 months, 89.5 months and 73.3 months for FNCLCC 1–3, respectively ($p = 0.005$) (HR 2.047, CI: 1.27:3.29).

**Table 1.** Univariate Analysis for Factors Impacting Progression Free Survival in recurrent patients only.

| Variable | | Univariate PFS Analysis | | | |
|---|---|---|---|---|---|
| | | Kaplan Meier | | Cox Regression | |
| | | Mean | *p*-Value | HR (95% CI) | *p*-Value |
| Gender | Female<br>Male | 72.885<br>42.610 | 0.343 | 1.639<br>(0.585–4.596) | 0.348 |
| FNCLCC Grade | FNCLCC (1)<br>FNCLCC (2)<br>FNCLCC (3) | 74.911<br>42.066<br>47.243 | 0.741 | 1.166<br>(0.664–2.046) | 0.593 |
| Prior Surgery | No<br>Yes | 43.958<br>62.859 | 0.548 | 0.729<br>(0.259–2.050) | 0.549 |
| Outcome Prior Surgery | STR<br>GTR<br>Unknown | 69.377<br>51.137<br>69.377 | 0.970 | 0.954<br>(0.418–2.176) | 0.912 |
| Prior Radiation Treatment | No<br>Yes | 76.770<br>24.084 | 0.002 | 6.359<br>(1.751–23.087) | 0.005 |
| Prior Chemotherapy | No<br>Yes | 67.296<br>15.361 | 0.000 | 15.437<br>(2.694–88.454) | 0.002 |
| Anterior Fossa | No<br>Yes | 63.233<br>43.424 | 0.545 | 1.360<br>(0.501–3.691) | 0.547 |
| Middle Fossa | No<br>Yes | 92.800<br>39.358 | 0.086 | 3.040<br>(0.812–11.374) | 0.099 |
| Posterior Fossa | No<br>Yes | 55.076<br>– | – | – | – |
| Dural Involvement | No<br>Yes | 84.354<br>42.877 | 0.260 | 2.083<br>(0.567–7.643) | 0.269 |
| Brain Invasion | No<br>Yes | 66.085<br>39.349 | 0.249 | 1.898<br>(0.628–5.737) | 0.256 |
| Cavernous Sinus Involvement | No<br>Yes | 97.967<br>37.205 | 0.052 | 3.346<br>(0.924–12.120) | 0.066 |
| Cranial Nerve Involvement | No<br>Yes | 57.392<br>52.471 | 0.849 | 0.905<br>(0.322–2.543) | 0.850 |
| Resection Status | R2<br>R1<br>R0 | 42.066<br>31.361<br>59.393 | 0.610 | 0.663<br>(0.271–1.622) | 0.368 |

### 3.2. Histological Breakdown and Multimodality Treatment Strategies

Histological breakdown of the cohort by tissue of origin is shown in Table 2. Categorized by FNCLCC histological grade, 26 patients (51%) were FNCLCC grade 3, while 8 patients (15.7%) were FNCLCC grade 2 and 17 patients (33.3%) were FNCLCC grade 1. While all patients in this cohort underwent surgical resection, the indications for, and timing of, chemo and radiation therapy were tailored to the biologic grade of the tumor and contemporary treatment strategies. The neoadjuvant and adjuvant therapies within each FNCLCC cohort are shown in Figure 1. Amongst the entire cohort, previous radiation therapy prior to presentation was the most common contra-indication to not offering

radiation therapy as part of our treatment algorithm. Our current medical and surgical management algorithm is shown in Figure 2.

**Table 2.** Tumor histology by histologic subtype and FNCLCC tumor grade.

| Tumor Histology by Differentiation | FNCLCC Tumor Grade | | | |
| --- | --- | --- | --- | --- |
|  | FNCLCC (1) | FNCLCC (2) | FNCLCC (3) | Total |
| Skeletal muscle tumors | | | | |
| Rhabdomyosarcoma | 0 | 0 | 10 | 10 |
| Fibroblastic or myofibroblastic tumors | | | | |
| Fibrosarcoma | 3 | 1 | 0 | 4 |
| Hemangiopericytoma | 8 | 5 | 3 | 13 |
| Dermatofibrosarcoma Protuberans | 0 | 1 | 0 | 1 |
| Malignant Fibrous Histiocytoma | 0 | 1 | 4 | 5 |
| Vascular tumors | | | | |
| Angiosarcoma | 0 | 0 | 1 | 1 |
| Adipocytic tumors | | | | |
| Liposarcoma | 1 | 0 | 5 | 6 |
| Pleomorphic Liposarcoma | 0 | 0 | 0 | 0 |
| Synovial tissue tumors | | | | |
| Synovial Sarcoma | 1 | 0 | 2 | 3 |
| Nerve sheath tumors | | | | |
| Neurofibrosarcoma | 2 | 0 | 3 | 5 |
| Smooth muscle tumors | | | | |
| Leiomyosarcoma | 1 | 0 | 0 | 1 |
| Tumors of Uncertain Differentiation | | | | |
| Sarcoma [NOS] | 1 | 0 | 0 | 1 |
| Alveolar Soft Part Sarcoma | 0 | 0 | 1 | 1 |
| Total | 17 | 8 | 26 | 51 |

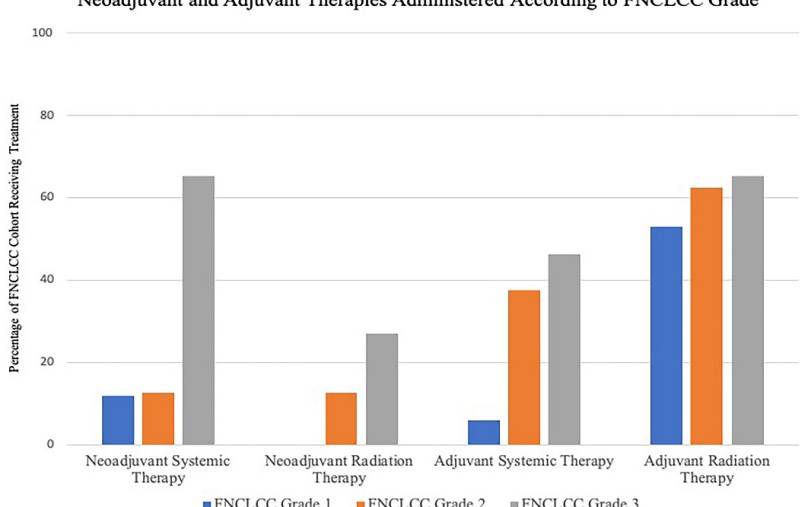

**Figure 1.** Neoadjuvant and Adjuvant Therapies Administered According to FNCLCC Grade. Y-Axis refers to percentage of patients in each FNCLCC cohort receiving treatment. Of the 17 FNCLCC grade 1 tumors, 2 patients (11.8%) underwent neoadjuvant chemotherapy. One patient (5.9%) received adjuvant chemotherapy and 9 (53%) received adjuvant radiotherapy. In the FNCLCC Grade 2 group, neoadjuvant radiotherapy was administered to one of 8 patients (12.5%) while 1 patient (12.5%) underwent neoadjuvant chemotherapy. In the adjuvant setting, 3 patients (37.5%) underwent chemotherapy, and 5 (62.5%) underwent radiotherapy. Seven of the 26 patients (26.9%) with FNCLCC grade 3 tumors received neoadjuvant radiotherapy and 17 (65.4%) received neoadjuvant chemotherapy. Twelve patients (46.2%) underwent adjuvant chemotherapy and 17 (65.4%) underwent adjuvant radiotherapy.

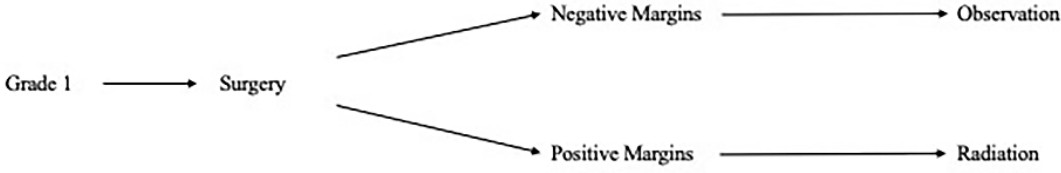

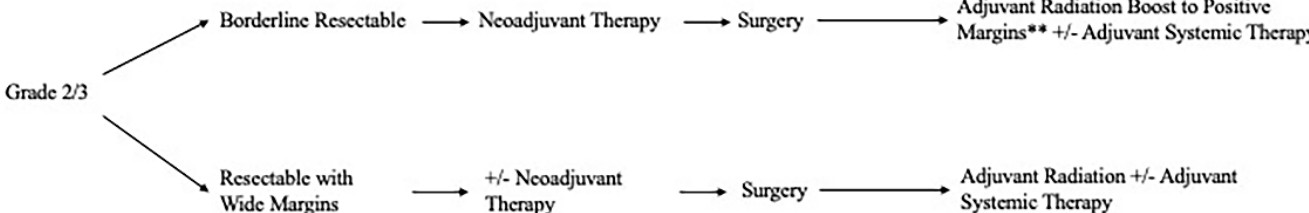

**Figure 2.** Current Multimodal Management Approach to Soft Tissue Sarcomas with Skull Base Involvement based on Histological Grade.

### 3.3. Tumor Extent and Surgical Strategies

The anatomic extent of the cohort is shown in Table 3. Extensive invasion into numerous anatomic compartments was noted with a higher predilection for disease in the middle fossa (68.6%). Reflective of the higher proportion of middle fossa involvement, the infratemporal fossa was involved in 47.1% of the cohort. While the origin of these tumors was primarily extradural, dural involvement was noted in 64.7% and cavernous sinus involvement in 37.3% of the cohort. The surgical strategies employed are also shown in Table 3. R0 resection was achieved in 80.4%, while R1 resection was achieved in 6.3% of the cohort. Statistical analysis demonstrated no difference between the negative and positive margin cohorts with regards to the anatomic extent of disease. Brain invasion was the only factor predictive of positive margin status ($p$ = 0.03). The duration between the first resection and the second typically ranged between 2–6 weeks.

**Table 3.** Summary of Tumor and Surgical Treatments.

| Characteristic | Subcategory | Total, $n$ = 51 (% of Cohort) |
|---|---|---|
| Skull base involvement | Anterior fossa | 16 (30.8) |
| | Middle fossa | 35 (67.3) |
| | Posterior fossa | 5 (9.8) |
| Subcranial involvement | Nasal cavity | 6 (11.5) |
| | Frontal sinus | 3 (5.8) |
| | Ethmoid sinus | 9 (17.3) |
| | Maxillary sinus | 14 (26.9) |
| | Orbit | 13 (25) |
| | Sphenoid sinus | 9 (17.3) |
| | Infratemporal fossa | 24 (46.2) |
| | Pterygopalatine fossa | 23 (44.2) |
| | Temporal bone | 25 (48.1) |
| | Clivus | 6 (11.5) |
| | Craniocervical junction | 3 (5.8) |

**Table 3.** *Cont.*

| Characteristic | Subcategory | Total, *n* = 51 (% of Cohort) |
|---|---|---|
| Extent of intracranial invasion | Bone | 47 (92.2) |
| | Dura | 33 (63.5) |
| | Brain | 16 (30.8) |
| | Cavernous sinus | 19 (36.5) |
| | Cranial nerves | 27 (53) |
| | Arterial | 8 (15.4) |
| | Venous | 23 (45) |
| Surgical Strategies | | |
| | Subtotal resection (STR) | 7 (13.5) |
| | Gross total resection (GTR) with positive margins | 3 (5.8) |
| | Gross total resection (GTR) with negative margins | 41 (80.3) |
| Surgical Reconstruction | | |
| | Primary closure | 37 (71.2) |
| | Rotational Flap | 1 (1.9) |
| | Free Flap Reconstruction | 13 (25.5) |
| Surgery Complications | | |
| Short Term | Airway compromise, epidural hematoma, post-op pneumocephalus with CSF leak | 3 (5.8) |
| Long Term | Ethmoid defect causing enophthalmos | 2 (3.8) |

*3.4. Recurrence Patterns and Predictors of Long-Term Outcomes*

With a median follow up of 65.6 months (range: 5–250 months) for the cohort, the median PFS was 263.4 months, while the 5- and 10-year PFS rates were 44% and 17%, respectively. The patterns of recurrence for the entire cohort, and then by FNCLCC grade, are shown in Table 4. It was evident that increasing FNCLCC grade was associated with an increasing risk of local and distant disease spread.

**Table 4.** Recurrence and disease status.

| | Overall Cohort | FNCLCC Grade 1 | FNCLCC Grade 2 | FNCLCC Grade 3 |
|---|---|---|---|---|
| | *N* = 51 Patients | | | |
| Mean PFS (months) | 153.788 | 181.320 | 117.895 | 91.885 |
| Recurrence | 17 | 5 | 1 | 11 |
| Local | 10 | 5 | 1 | 4 |
| Distant | 4 | 0 | 0 | 4 |
| Local and distant | 3 | 0 | 0 | 3 |
| Median disease specific survival (DSS) (months) | 94.5 | 170.7 | 89.2 | 73.2 |
| Status at Last Follow-Up | | | | |
| Alive | 22 | 13 | 3 | 6 |
| Dead | 29 | 4 | 5 | 20 |

The results of the univariate and multivariate analysis are shown in Tables 5 and 6. In the overall cohort, the ability to achieve an R0 resection was associated with a trend toward improved PFS and a reduced risk of recurrence over the study time period (HR 1.050, CI 0.483–2.283, *p* = 0.902). When assessing only radiation naïve patients, an R0 resection yielded a near significant improvement in mean PFS (174 months vs. 87 months, *p* = 0.06) (Figure 3). Otherwise, with regards to PFS analysis, prior radiation therapy (HR 3.6, CI 1.21–10.9, *p* = 0.021) and cavernous sinus involvement (HR 11.39, CI 3.16–41.09, *p* = 0.00) negatively impacted tumor control outcomes.

**Table 5.** Univariate Analysis for Factors Impacting Progression Free Survival.

| Variable | | Univariate PFS Analysis | | | |
| --- | --- | --- | --- | --- | --- |
| | | Kaplan Meier | | Cox Regression | |
| | | Mean | *p*-Value | HR (95% CI) | *p*-Value |
| Gender | Female<br>Male | 162.406<br>112.584 | 0.723 | 1.201<br>(0.434–3.32) | 0.724 |
| FNCLCC Grade | FNCLCC (1)<br>FNCLCC (2)<br>FNCLCC (3) | 181.320<br>117.895<br>91.885 | 0.258 | 1.516<br>(0.832–2.760) | 0.174 |
| Prior Surgery | No<br>Yes | 108.337<br>161.593 | 0.532 | 0.730<br>(0.271–1.967) | 0.533 |
| Outcome Prior Surgery | STR<br>GTR<br>Unknown | 181.861<br>85.798<br>121.491 | 0.704 | 1.085<br>(0.503–2.340) | 0.836 |
| Prior Radiation Treatment | No<br>Yes | 172.890<br>49.991 | 0.014 | 3.645<br>(1.213–10.953) | 0.021 |
| Prior Chemotherapy | No<br>Yes | 159.495<br>93.530 | 0.174 | 2.166<br>(0.692–6.748) | 0.184 |
| Anterior Fossa | No<br>Yes | 159.342<br>138.744 | 0.621 | 1.278<br>(0.482–3.386) | 0.622 |
| Middle Fossa | No<br>Yes | 165.214<br>147.249 | 0.442 | 1.505<br>(0.527–4.298) | 0.445 |
| Posterior Fossa | No<br>Yes | –<br>– | 0.176 | 0.042<br>(0.0–50.757) | 0.382 |
| Dural Involvement | No<br>Yes | 174.260<br>106.761 | 0.361 | 1.703<br>(0.538–5.393) | 0.366 |
| Brain Invasion | No<br>Yes | 169.577<br>80.300 | 0.132 | 2.135<br>(0.779–5.855) | 0.141 |
| Cavernous Sinus Involvement | No<br>Yes | 204.512<br>49.515 | 0.000 | 11.393<br>(3.159–41.095) | 0.000 |
| Cranial Nerve Involvement | No<br>Yes | 155.029<br>111.816 | 0.774 | 0.866<br>(0.324–2.316) | 0.774 |
| Resection Status | R2<br>R1<br>R0 | 117.895<br>36.896<br>155.741 | 0.105 | 1.050<br>(0.483–2.283) | 0.902 |

**Table 6.** Multivariate Analysis for Factors Impacting Progression Free Survival and overall survival.

| Variable | | Multivariate OS Analysis Cox Regression | | Multivariate PFS Analysis Cox Regression | |
| --- | --- | --- | --- | --- | --- |
| | | HR (95% CI) | *p*-Value | HR (95% CI) | *p*-Value |
| Prior Radiation Treatment | **No**<br>**Yes** | 2.477<br>(1.066–5.759) | 0.035 | 5.734<br>(0.774–19.530) | 0.005 |
| Prior Chemotherapy | **No**<br>**Yes**<br>**Yes** | 2.181<br>(0.736–6.459) | 0.159 | 2.686<br>(0.766–9.419) | 0.123 |
| Posterior Fossa | **No**<br>**Yes**<br>**Yes** | 0.742<br>(0.134–4.108) | 0.732 | 0.000 | 0.985 |

**Table 6.** *Cont.*

| Variable | | Multivariate OS Analysis Cox Regression | | Multivariate PFS Analysis Cox Regression | |
|---|---|---|---|---|---|
| Brain Invasion | No | 2.981 | 0.027 | 0.778 | 0.658 |
| | Yes | (1.130–7.861) | | (0.256–2.364) | |
| Cavernous Sinus Involvement | No | 1.872 | 0.161 | 18.712 | 0.000 |
| | Yes | (0.780–4.494) | | (3.960–88.412) | |
| Resection Status | R2 | 0.804 | 0.434 | 0.774 | 0.574 |
| | R1 | (0.465–1.390) | | (0.316–1.891) | |
| | R0 | | | | |

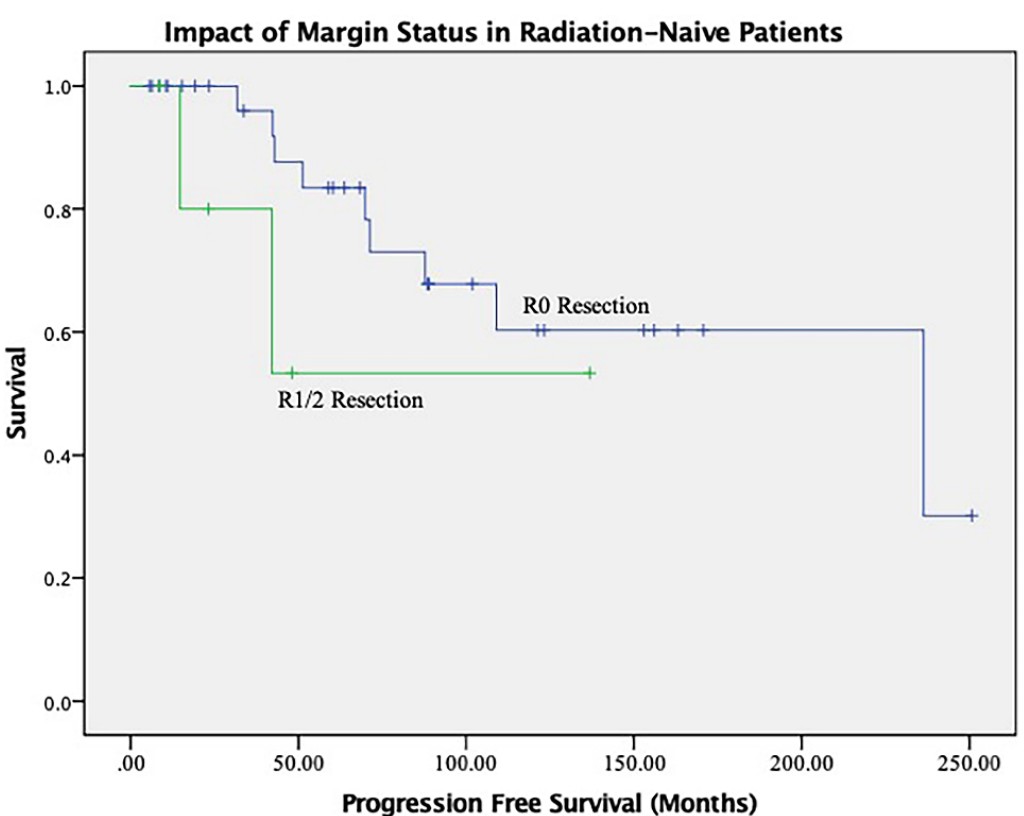

**Figure 3.** Impact of R0 Resection on Tumor Control in Radiation Naïve Patients.

## 4. Discussion

In the management of patients with T4 stage head and neck STS with skull base invasion, the data shows that the most common cause of treatment failure was local recurrence. Prior radiation treatment to multimodal treatment at our institution and cavernous sinus involvement were independent predictors of poor PFS, while R0 resection improved tumor control rates in radiation naïve patients, despite the surgical challenges in T4-stage patients.

In line with outcomes at other disease sites, this study demonstrated local recurrence to be the most common pattern of disease failure and risk of systemic metastasis that appeared to correlate with biological grade. This finding highlighted the importance of histological grade in predicting clinical outcomes and the goals of treatment. Several histologic grading systems have been reported to circumvent the challenges in managing a heterogeneous group of tumors that span different histologic cell types. The French grading system and the National Cancer Institute are the two most notable, both of which are 3-tiered systems. The primary difference is the inclusion of anatomic location, which does not necessarily account for inherent challenges with skull base locations. A comparative analysis of 410 STS at all

sites validated the prognostic value of both systems in predicting overall survival, local recurrence, and risk of metastases [11]. Ultimately, biologic grade drives the indications for radiation therapy and chemotherapy in conjunction with surgery.

For resectable disease, surgery is a cornerstone in the management paradigm. The gold standard in extremity and visceral sarcoma surgery is considered to be en bloc resection with wide margins (reported to be up to 2.5 cm in some studies). Despite the inherent challenges in the skull base and the anatomically extensive tumors in this cohort, the ability to achieve a negative margin resection confers advantages with regard to local and distant disease control. It is important to note that higher biological grade did not correlate with the inability to achieve negative margins. Although the majority of patients in this cohort had FNCLCC grade 3 disease, the rate of GTR with negative margins was 86.1% in this subset of patients with grade 3 disease, higher than the same rates in patients with grade 2 (72.2%) and grade 1 (83.3%) disease. An R0 resection yielded significantly improved tumor control rates in radiation-naïve patients in our study. The published data on the impact of margin status in soft tissue sarcomas is mixed, with some studies on sarcomas of the head and neck indicating that margin status does not significantly impact outcomes [12,13]. However, in their International Collaborative of 146 anterior skull base sarcomas (encompassing both bone and soft tissue sarcomas) Gil et al. found that a positive surgical margin was the only independent predictor of poor disease-specific survival [14]. These discordant findings regarding the impact of margin status on survival are likely confounded by the inclusion of bony sarcomas in some reported series in addition to the inconsistent reporting of biological grade and heterogeneous multimodal management strategies. Additionally, the risk of metastatic disease and the availability of multimodal treatment options are also likely to influence survival beyond margin status.

This study also highlights the challenges in managing soft tissue sarcomas, not only with regards to their spectrum of biological behavior, but also the indications for radiation and systemic therapy and their timing relative to surgery [15]. Over the years, there have been controversies about the use of adjuvant radiotherapy for the treatment of tumors, especially after R0 resection [14]. However, adjuvant radiotherapy should be considered for intermediate or high-grade tumors and positive surgical margins [8,12,14]. In a cohort of 1093 extremities and trunk wall STS patients, Jebsen et al. demonstrated a significant impact with radiotherapy on tumor control in high-grade patients, even after a wide margin resection [16]. Adjuvant radiotherapy improved local control rate, which was important, considering that failures are most commonly local in [14]. This was also demonstrated in this study. Adjuvant radiotherapy has become a cornerstone of treatment in STS of the skull base over the last decade [8]. Another debate regarding radiation pertains to timing relative to surgery. With disease sites outside the skull base, preoperative treatment can require smaller fields and require lower doses, potentially translating into lower late treatment toxicity [17–20]. This is balanced by concerns with wound healing complications that can be mitigated by postoperative delivery. A randomized trial allocating patients to preoperative or postoperative radiotherapy demonstrated a significantly higher rate of wound complications with preoperative treatment [21]. Unfortunately, there is no clear consensus in the literature regarding an optimal treatment sequence. In our own practice, we favor neoadjuvant radiation for high-grade, borderline resectable sarcomas, with the option for a radiation boost in the adjuvant setting to positive margins.

The increasing risk of metastatic relapse with higher grade histology underscores the need for systemic therapies. Due to conflicting data from large studies, current clinical practice guidelines do not suggest adjuvant systemic therapy as standard practice. Given the poor response of STS to conventional anthracycline-based chemotherapy, there has been a push over the last decade towards differing systemic therapy regimens [15]. A recent trial comparing epirubicin/ifosfamide versus histology-specific chemotherapy in the neoadjuvant setting in high-risk patients demonstrated a significant benefit in tumor control and survival rates with epirubicin plus ifosfamide [22]. This study has ultimately been viewed as demonstrating randomized evidence in support of neoadjuvant anthracycline

plus ifosfamide in high-risk patients having high-grade histology, and deep-seated tumors and/or tumor size > 5 cm. In our own cohort, neoadjuvant therapy was employed in 38% of the cohort in which radiographic response was noted in 65%. Of note, this strategy of neoadjuvant therapy allows for direct and early feedback regarding efficacy, based on radiographic and histologic results, ultimately guiding adjuvant therapy where ineffective therapies can be abandoned early.

Our current treatment algorithm for skull base STS is shown in Figure 2. While observations from larger studies at other disease sites and NCCN recommendations serve as a framework, factors unique to the skull base must also be considered. As new targeted and immunotherapy options are assessed, this treatment algorithm will certainly be reassessed.

This is a retrospective cohort study from a single institutional database, with the selection biases associated with the nature of these studies. There is also a small number of patients, many of whom received prior treatment before arrival at our institution, limiting the generalizability of these results. Additionally, the numerous histologic diagnoses that comprise this category of sarcomas results in a seemingly biologically heterogeneous cohort of patients. This is a similar challenge even noted with large clinical trials performed on STS at other anatomic sites. In order to circumvent this, similar to previously published clinical trials, we used the FNCLCC grading system to categorize this cohort. Given the rarity of this disease and the lack of published data to guide treatment, these results can be used to inform surgical decision-making and counseling for patients with STS of the skull base.

## 5. Conclusions

Despite the prognostic and management implications of T4-stage head and neck sarcomas with skull base involvement, for radiation naïve patients, a margin negative resection is still impactful and the ability to do so has a long-term impact on tumor control rates. Beyond anatomic staging, biologic staging with the FNCLCC system is an important consideration regarding the neoadjuvant and adjuvant treatment strategies in light of its value in predicting the pattern of recurrence and long-term outcomes. This study adds to the limited results reported in the literature regarding this high-risk set of diseases. Future work needs to be done to help elucidate the role of neoadjuvant radiotherapy in the skull base and optimal management strategies for post-radiation disease recurrence.

**Author Contributions:** Conceptualization, S.M.R.; methodology, S.M.R., A.H. and I.E.; software, A.H. and I.E.; validation, All authors.; formal analysis, A.H.; investigation, S.M.R., A.H., D.B., S.Y.S. and E.Y.H.; resources, M.E.K.; data curation, A.H.; writing—original draft preparation, A.H. and S.M.R. writing—review and editing, S.M.R., A.H. and F.D.; visualization, A.H.; supervision, S.M.R.; project administration, S.M.R.; funding acquisition, N/A. All authors have read and agreed to the published version of the manuscript.

**Funding:** This research received no external funding.

**Institutional Review Board Statement:** This study was done using a protocol not requiring patient consent, the Institutional Review Boards' committee in our institution reviewed the study and deemed it in compliance with institutional regulations regarding the study of human subjects. The protocol was institutional review-board approved and met all HIPPA standards.

**Informed Consent Statement:** Not applicable.

**Data Availability Statement:** The data presented in this study are available on request from the corresponding author.

**Conflicts of Interest:** Authors declare no conflict of interest.

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
