# Peer review of "Soft Tissue Sarcomas of the Head and Neck Region with Skull Base/Intracranial Invasion: Review of Surgical Outcomes and Multimodal Treatment Strategies: A Retrospective Case Series"

_curroncol, doi:10.3390/curroncol29090514_

Round 1

Reviewer 1 Report

This retrospective review analyses progression free survival and recurrence rate (local and/or systemic) of 51 patients affected by soft tissue sarcomas of the head and neck region, surgically treated in a high-volume centre in a period of 23 years. Only patients with local disease without systemic or nodal metastases were included. Tumor histologic subtype and grade (FNCLCC) and extent of resection are considered main variables, but adjuvant and neoadjuvant therapies, tailored according to FNCLCC grade, are reported and discussed. Discussion and conclusions are clearly exposed and adequate and the Authors themselves underline limitations of the study.

I agree that “given the rarity of this disease and the lack of published data to guide treatment, these results can be used to inform surgical decision-making and counseling for patients with STS of the skull base. Moreover, this paper may stimulate multicentre reviews. In my opinion, this paper may be accepted for publication after minor text revision.

Author Response

We thank the reviewer for his/her comments.

Reviewer 2 Report

This is an interesting study about soft tissue sarcomas of the head and neck region with skull base/intracranial invasion.

The paper is well written. However, some issues remain.

Acronyms should be explained at their first appearance both in the abstract and the text.

The authors should add criteria for neoadjuvant and adjuvant radiotherapy and/or chemotherapy.

Overall survival should be analyzed and added to the paper.

Please describe how much time there was between the two surgical procedures in some patients.

A multivariate analysis may be helpful.

Moreover, patients with recurrent disease must be analyzed separately.

Author Response

Reviewer: -

This is an interesting study about soft tissue sarcomas of the head and neck region with skull base/intracranial invasion.

The paper is well written. However, some issues remain.

Acronyms should be explained at their first appearance both in the abstract and the text.

Authors: - we thank the reviewer pointing this out. We have explained the acronym in the abstract and the main text.

The authors should add criteria for neoadjuvant and adjuvant radiotherapy and/or chemotherapy.

Authors: - we have explained the criteria of choosing chemoradiation in relation to the histological features of each tumor in the discussion section lines 224-237.

Overall survival should be analyzed and added to the paper.

Authors: - we have included the overall survival data in the results section line 105-108.

Please describe how much time there was between the two surgical procedures in some patients.

Authors: - this duration typically ranged between 2 – 6 weeks.   

A multivariate analysis may be helpful.

Authors: - we thank the reviewer for this comment. We would like to point out that only “Prior Radiation Treatment” “Cavernous Sinus Involvement” were the only significant predictors of the PFS in the univariate analysis. Based on that we did not believe that a multivariate analysis will be of benefit in this study.

Moreover, patients with recurrent disease must be analyzed separately.

Authors: - we described analyzed in detail the patients with recurrent disease and dedicated table 3 for these analyses. Analyses includes PFS and DSS.

Round 2

Reviewer 2 Report

The criteria for neoadjuvant and adjuvant radiotherapy and/or chemotherapy must be added in the methods section, not only in the discussion.

The time between the two surgical procedures should be added in the paper.

I think that a multivariate analysis should be performed.

The 47.1% of patients who had recurrent disease at presentation must be analysed separately.

Author Response

The criteria for neoadjuvant and adjuvant radiotherapy and/or chemotherapy must be added in the methods section, not only in the discussion.

  • We referred to it again in the methods. Additionally we need to point the attention of the reviewer towards the management Algorithm including neo and adjuvant therapies are described in detail in figure 3.

The time between the two surgical procedures should be added in the paper.

  • We have added the duration to the text.

I think that a multivariate analysis should be performed.

- we have performed a multivariate analysis and added to a new table (table 5)

-

The 47.1% of patients who had recurrent disease at presentation must be analysed separately.

  • we have performed a separate analysis to this subgroup and added to a new table (table 6)

Round 3

Reviewer 2 Report

Thank you for improving the manuscript.